# Can Forest Trees Cope with Climate Change?—Effects of DNA Methylation on Gene Expression and Adaptation to Environmental Change

**DOI:** 10.3390/ijms222413524

**Published:** 2021-12-16

**Authors:** Ewelina A. Klupczyńska, Ewelina Ratajczak

**Affiliations:** Institute of Dendrology, Polish Academy of Sciences, Parkowa 5, 62-035 Kórnik, Poland; eratajcz@man.poznan.pl

**Keywords:** epigenetics, DNA methylation, plants, forest trees, climate change, reprogramming genes, editing epigenome

## Abstract

Epigenetic modifications, including chromatin modifications and DNA methylation, play key roles in regulating gene expression in both plants and animals. Transmission of epigenetic markers is important for some genes to maintain specific expression patterns and preserve the status quo of the cell. This article provides a review of existing research and the current state of knowledge about DNA methylation in trees in the context of global climate change, along with references to the potential of epigenome editing tools and the possibility of their use for forest tree research. Epigenetic modifications, including DNA methylation, are involved in evolutionary processes, developmental processes, and environmental interactions. Thus, the implications of epigenetics are important for adaptation and phenotypic plasticity because they provide the potential for tree conservation in forest ecosystems exposed to adverse conditions resulting from global warming and regional climate fluctuations.

## 1. Introduction

The discovery of epigenetic modifications of DNA and DNA-related modifications of chromatin proteins is of great scientific importance because it provides new directions for research and greatly expands the prospects of experimental biology. As research has progressed, it has become clear that genes are not the sole determinant of heritability and variation in traits or the development of organisms. Epigenetics has shown that a second, highly important stimulator is the environment in which organisms live. This is the case for both plant and animal kingdoms. According to the latest scientific knowledge, ‘epigenetics’ refers to the science that deals with the study of modifications occurring in the genome that regulates the level of gene expression, without prior changes in the nucleotide sequence. Epigenetic modifications are stable, reversible changes in DNA or histones that can be inherited but are also dependent on external stimuli (e.g., environmental and endogenous factors, such as aging).

DNA methylation is now well understood. Methylation occurs in both plants and animals and is an important epigenetic modification due to its role in gene regulation, transposable element (TE) silencing, chromosomal interactions, and stability of the plant genome [1]. In addition to methylation, other epigenetic modifications include chromatin remodeling by associated proteins, histone modifications, RNA interference, or regulation by non-coding RNA (ncRNA), otherwise known as regulatory RNA. Modifications of DNA and histone proteins remodel chromatin, i.e., change its structure, which, among other things, alters the activity of genes and transposons and aids in DNA repair. Changes in the way DNA is packaged offer the possibility to control its reading. The finding that epigenetic modifications can control the genome of eukaryotic organisms is a groundbreaking one [2,3]. Following this discovery, it also became clear that epigenetics plays an extremely important role in maintaining genome stability and is involved in crucial biological processes.

In plants, DNA methylation is involved in regulating many biological processes that determine their emergence and development, and largely determines their rapid response to environmental changes, supporting adaptation processes. Global climate change negatively impacts some forest ecosystems and forest tree species, but these are not well understood. The broadest possible knowledge, at the level of the genome and molecular mechanisms of the adaptive potential of forest trees (including seed development), would allow increasingly effective monitoring of the status of tree populations of threatened species and provide further opportunities for forest conservation action.

## 2. Effects of DNA Methylation of Forest Trees on Gene Expression and Climate Adaptation

### 2.1. DNA Methylation in Plants

Plants are very specific organisms with unique abilities that lead them to maximize their potential. This feature is due to the non-mobility of plants. Admittedly, plants live in one specific environment but are exposed to a wide range of fluctuations, including climatic fluctuations, which can permanently or temporarily change their living environment. Consequently, plants continually produce various evolutionary biological adaptations. It is known that environmental conditions can also induce gene activity through epigenetic processes. Plants have developed a number of extremely complex epigenetic mechanisms that they use to control and regulate the genome. Genome potential control is possible through changes in chromatin structure. Epigenetic modifications alter the ability of genes to be expressed and directly coordinate with primary metabolism, which regulates plant growth and development. Epigenetic processes affect both the phenotype and fitness of plants and contribute to their ability to colonize and evolve in changing environments [1,4,5]. Environmentally induced epigenetic modifications represent an important adaptive strategy. In this context, the study of epigenetic modifications in plants is extremely interesting and demonstrates that forest trees have enormous adaptive potential.

DNA methylation is a post-replicative enzymatic modification of DNA. It is a stable process leading to the silencing of gene expression and resulting in the covalent attachment of so-called methyl groups (one carbon atom joined to three hydrogen atoms, -CH3) to the nitrogenous bases of nucleotides (cytosine and adenine). In higher eukaryotes, methyl groups attach to a carbon atom located at position five of the cytosine ring of the DNA double helix, forming C5-methylcytosine (m5C). They can also attach to the amino group of cytosine (N4, m4C). In some higher plants, N6-methyladenine (m6A) is also formed as a methylation effect [6] (Figure 1).

In plants, DNA methylation occurs in all contexts of the CpG, CpHpG, and CpHpH dinucleotide sequences (H represents any A, T, or C nucleotide, except G) located in a linear 5′ to 3′ DNA sequence, with the cytosine and guanine nucleotide adjacent or one base apart [7,8,9]. CpG dinucleotides are randomly distributed throughout the genome and are mostly methylated. However, they also form sites with lower levels of methylation where CpG sequences are highly concentrated; these are referred to as CpG islands (CGIs). CGIs are regions more than 200 bp long [10], and CpG sites in CGI are mostly unmethylated, thus avoiding mutational deamination of C5-methylcytosine (m5C) to thymine. CGIs often overlap with transcription start sites (TSSs), and their hypermethylation may be associated with transcriptional repression [11]. It was previously thought that methylation could only silence genes at CGI sites. However, this dogma was overturned by the recent scientific reports that have documented that CpG islands are not needed to turn off genes by methylation [12].

Chromatin can be decondensed (open chromatin or euchromatin), enriched in genes and promoting transcription, or condensed (closed chromatin or heterochromatin), enriched in repetitive sequences and silenced sequences [1]. Changes in chromatin are most often introduced by small RNAs (sRNAs) and longer non-coding RNAs (lnRNAs), which are conduits for mechanisms of DNA methylation, chromatin modification, or transcript degradation/amplification [13]. Reversible changes in chromatin structure, such as cytosine methylation or histone modifications, directly affect the transcriptional efficiency of genes [14,15].

The maintenance of methylation is controlled by several pathways (Figure 2). CpG methylation is maintained by DNA methyltransferase 1 (MET1). In addition, more recently, MET1 has been assigned a broader function, as studies show that it is also required for the establishment of CpHpH methylation [16].

Maintenance of DNA methylation in the CpHpG context requires chromomethylase 3 (CMT3) activity. Methylation in the CpHpH context requires chromomethylase 2 (CMT2) activity [17,18], and a large amount of CpHpH methylation is maintained by domains rearranged methyltransferase 2 (DRM2) in the RNA-dependent DNA methylation (RdDM) pathway [15,19,20,21], which is also responsible for de novo methylation in all three sequence contexts [8,9,18]. CpHpH methylation is asymmetric, meaning methylation will be lost in one progeny strand. Chromomethylases contain both a chromodomain and a DNA methyltransferase domain and interact with some proteins (suppressor of variegation su(var) homolog, SUVH) to ensure proper deposition of histone H3K9 (H3K9me2) methylation, as well as CpHpG or CpHpH in transposable elements (TEs) [20]. SUVH proteins are essential for accessing the regulatory mechanisms of genes located in close proliferating transposable elements (TEs) [20]. DRM2 and MET1 proteins share significant homology with mammalian methyltransferases (DNMT3 and DNMT1). The CMT3 protein is unique to plants and belongs to a family of chromomethylases that are both “readers” of histone methylation and “writers” of DNA methylation [20,22]. Most methylation in plants occurs in transposable elements (TEs), but also in the bodies of active genes where it is restricted to the CpG context [9,18]. The first step in de novo DNA methylation pattern formation is the RNA-dependent DNA methylation pathway, which relies on specialized, plant-specific RNA polymerases POL IV or POL V (POL VI is also specific for grasses) [20,22] (Figure 3).

Important functions of the RdDM pathway include the control of repeats in heterochromatic regions and dispersed transposons, as well as related sequences in euchromatic regions [8,24]. Transcriptional repression of actively proliferating transposons for genome defense and activation of the parent-of-origin expression of specific genes in reproductive tissues are also included [25]. Pol IV and Pol V polymerases uniquely contribute to epigenetic regulation by producing silencing-related non-coding transcripts. The silencing pathway through Pol IV and Pol V activity can be counteracted by active demethylation, thereby creating epigenetic flexibility that is important for environmental adaptation [13].

### 2.2. Forest Trees—Ecosystems Important to Humans

Forests cover approximately 31% of the world’s land area, or 4 billion hectares (http://www.fao.org/state-of-forests/en, accessed on 19 July 2021). Almost half of these are intact (natural forests), and more than one-third are naturally regenerating forests of native species where there are no traces of human activities and natural ecological processes are undisturbed (primary forests). Forests are dynamic ecosystems with high environmental, social, and economic importance [1]. However, the global changes currently occurring on Earth, such as desertification, insect invasion, abiotic stresses, deforestation, degradation, and climate change, pose a significant threat to the condition of forests.

In 2017, a database on forests and forest-forming species of the world was established [26]. GlobalTreeSearch is the first global, authoritative list of tree species created through the multidisciplinary work of many organizations and associated with forest scientists [26]. One of the goals of GlobalTreeSearch is to protect forests.

Other global organizations have also become involved in addressing the threat to forests from global change, including the United Nations, which has created a forest aid plan—The United Nations Strategic Plan for Forests 2017–2030 (UNSPF) (https://www.un.org/esa/forests/wp-content/uploads/2016/12/UNSPF_AdvUnedited.pdf, accessed on 19 July 2021). The organization recognizes the need for global cooperation and coordination for sustainable forest development and creating more resilient and adaptable forest communities.

Forests perform important environmental functions, such as soil and water conservation, biodiversity protection, and the production of valuable raw materials and food for humans. Forest genetic resources are important for both adaptation processes, tree evolution, and forest biotopes, and for improving tree resilience and productivity. Monitoring forests and their genetic resources is needed now more than ever, at a time when the world is increasingly facing challenges from land-use change and climate change, among other factors that are causing loss of forest cover and forest biodiversity. Forests are among the world’s most productive terrestrial ecosystems and are essential to life on Earth, so it is paramount that similar initiatives such as GlobalTreeSearch are sustained with the comprehensive benefits of forest conservation in mind: social, environmental, and cultural for current and future generations. In contrast, limited information about the adaptive capacity of forest trees reduces the ability of many countries and the international community to develop appropriate tools to protect them. The last two decades have been a field of research and development for the conservation of forest genetic diversity. The importance of using appropriate, matching, diverse, and improved germplasm in forestry systems and the need for appropriate seed and seedling production systems are now being appreciated more. Ex-situ conservation ensures the survival of genetic resources [27,28,29,30]; in the case of forest genetic resources, this involves the storage of seeds, ensuring that they remain in suitable, species-specific conditions [31,32,33,34,35,36,37,38,39,40,41]. Skillful, effective seed storage provides a stable source of genetic diversity of forest trees needed for forest conservation. Plant conservation research in recent years has led to innovations in seed storage, e.g., diagnosing tree seed behavior in storage, increasing tree seed longevity in dry storage, and improving storage biotechnology [35,42,43,44,45]. To improve the ex-situ conservation of forest genetic resources using seed storage in gene banks, research on proper seed handling and identification of seed behavior during storage must continue persistently to further develop seed storage techniques (e.g., cryopreservation of seeds of forest tree species). Information regarding seed biology of forest tree species is limited [32,46,47,48,49] and available resources are scattered, thus making it all the more important to expand knowledge in this area. This is even more important as there are studies also confirming the association of epigenetic processes with seed storage [38,50]. It is also essential to target species-based conservation to maintain as much intraspecific high-priority forest tree species or endangered species tree diversity as possible. Such efforts are mostly observed in Europe and North America, such as *Picea abies* in Finland [51,52,53,54]. In tropical countries, the problem includes *Pinus merkusii* Jungh. and de Vriese in Asia [55,56,57], and *Terminalia richii* A. Gray and *Manilkara samoensis* H.J. Lam and B. Meeuse in Samoa [58].

The possibility that the plants themselves, including trees, have to thrive and create an ideal habitat for themselves in terms of adapting to any stress conditions relates to maximizing their adaptability and plasticity. Because of their longevity and exposure to large seasonal changes, trees and perennial woody plants, especially from temperate and boreal regions [4], have evolved phenotypic modification systems to tolerate changes in climatic conditions. Tree adaptation to the surrounding environment is based on the natural evolution of biological mechanisms (including epigenetics) that lead to the development of plant tolerance and resilience and the avoidance of environmental constraints [4,5]. Hence, there is a wide range of plant tolerances to edaphic and climatic conditions, from northern subarctic to subtropical zones. Among trees, conifers are considered the most adapted plants with the greatest adaptive potential [4]. They are also among the longest-lived terrestrial organisms. However, accelerating climate change, with increasing temperatures and decreasing precipitation, is challenging for all plants, including those with the greatest adaptive capacity. Although trees possess adaptive genomes and excellent regulatory mechanisms (genetic and epigenetic) whereby changes in gene activity occur in a plastic manner that allows them to survive and reproduce successfully in changing environments, their future is not known or predictable in terms of their ability to adapt to ongoing changes. Therefore, there is a need for further intensive research, especially in the field of epigenomics of forest trees, which is the least known area among plants and, in light of recent scientific reports, seems to be one of the key issues surrounding climate change.

The vast majority of studies devoted to plant epigenetics focus only on model herbaceous plants, such as *Arabidopsis thaliana* L. [18,59,60,61,62,63,64,65,66,67]. Studies using Arabidopsis also include work on global climate change [68,69,70,71,72,73]. Similarly, issues concerning the role of DNA methylation itself in the ability of plants to acclimate and adapt to changing environmental conditions or stress memory have also recently been quite extensively described [13,45,63,74,75,76].

In contrast, little research has been conducted on long-lived woody plants, omitting useful trees. Despite the growing climate problem, in recent years, there have been few studies on forest tree epigenetics in the context of stricte climate change [5,77,78,79,80]. As can be seen from the herbaceous plant research reports cited above, we have research tools that can expand our knowledge of trees and forest ecosystems. Therefore, there is still much work required to discover the mechanisms responsible for adaptation and all the environmental processes involved in forest trees.

Forest trees are globally dispersed, modular organisms with a complex life cycle. They are subjected to multiple environmental pressures during their long lives. Tree populations that survive environmental changes are the result of complex, interacting, and advanced evolutionary mechanisms, such as migration, adaptation, and phenotypic plasticity [1]. Trees, as long-lived organisms, may specifically use epigenetics to facilitate phenotypic modifications in response to environmental change [1]. Identifying genomic loci that undergo epigenetic changes in response to environmental conditions is an important research goal. By expanding this knowledge, it will become possible to understand the processes involved in epigenetic adaptation, that is, how epigenetic modifications heritably alter a plant’s gene expression and thereby refine its responses to environmental stimuli and its ability to survive under altered conditions [13].

### 2.3. Effects of DNA Methylation on Adaptations of Forest Trees

Epigenetic studies of forest trees have been significantly accelerated with the sequencing of the first tree genomes of *Populus trichocarpa* [81], *Picea abies* (L.) H. Karst. [82], and *Picea glauca* (Moench) Voss [83], or *Eucalyptus grandis* W. Hill [84] and *Pinus taeda* L. [85]. Today, there is a much-expanded forest tree genome database, with species such as *Pinus lambertiana* Douglas [86], *Ginkgo biloba* L. [87], *Fraxinus excelsior* L. [88], *Pseudotsuga menziesii* (Mirbel) Franco [89], *Betula pendula* Roth. [90], *Larix sibirica* Ledeb. [91], *Fagus sylvatica* L. [92], *Abies alba* Mill. [93], and *Eucalyptus pauciflora* Sieber ex Spreng. [94]. The more knowledge we acquire about forest tree genes, the better we can learn about regulatory epigenetic mechanisms. DNA methylation in the regulation of gene expression in tree responses to environmental stimuli has been widely studied (abiotic stress), including in droughts [38,95,96,97,98], heavy metals [99], extreme temperatures [100,101,102], and salt stress [103,104,105]. These changes can occur at the genome level [95,106]. In most cases, global demethylation of genomic DNA occurs in response to abiotic stress, but whether the same is happening in response to environmental climate change is not known. Methylation is known to play an important role in fine-tuning gene expression during plant development, as well as in response to the environment, enabling relatively rapid adaptation to new conditions without altering the DNA sequence [103]. Evidence for the involvement of methylation in environmental adaptation is also provided by a study of 1001 Arabidopsis DNA methylomes described in The 1001 Epigenomes Project [64]. These studies show that methylation levels within transposable elements positively correlate with latitude and precipitation, and negatively correlate with warmer temperatures. The question that arises in this context is whether plants, including forest trees, respond to climate change in the same way.

The methods used to analyze DNA methylation are often based on endonuclease digestion, affinity enrichment, and bisulfite conversion [107]. Common methods for studying cytosine methylation in DNA include, for example, the methylation-sensitive amplified polymorphism (MSAP) technique, genome-wide DNA methylation status analysis based on high-throughput Methyl Sensitive Deamination Adjacent to RNA modification Targets sequencing (MS-DART-seq), and sequencing or reduced-representation bisulfite sequencing (RRBS) or whole-genome bisulfite sequencing (WGBS)—which is another technique that enables genome-wide DNA methylation analysis with single CpG resolution [108]. WGBS and RRBS are highly compatible with each other and require conversion of genomic DNA with sodium bisulfite prior to sequencing on an NGS platform. These methods are used to study plant DNA methylation associated with factors such as development, transformation, heterosis, abiotic stresses, or pathogen interactions, among others [109].

DNA methylation is essential for plant embryogenesis and seed development. Abnormal methylation in the embryo causes defects in embryogenesis, such as impaired cell division, aberration of the embryo apical domain, and reduced viability [100,110]. The environment of the parents during reproduction also affects offspring performance. An example is the Norway spruce *Picea abies* (L.) H. Karst seedling, which “remembers” the temperatures and photoperiod that prevailed during their embryonic life and seed maturation. This memory affects climatic adaptation in this species and is an epigenetic phenomenon [100]. The existence of epigenetic memory in spruce may also explain the adaptive skills and rapid acclimatization of the spruce of Central European provenance in Norway [111]. The researchers suspect that changes in the mRNA of genetically identical, somatic spruce embryos during morphogenesis, under different temperature variants, may be related to chromatin modifications. The significant role of DNA methylation and histone and small RNA (sRNA) methylation in the formation of epigenetic memory in this species may also be indicated by the different expression of epigenetic regulators, variable under different conditions for epitopes [112,113]. The observation of global climate change makes the phenomenon of epigenetic memory during embryogenesis scientifically significant, as well as practically important in the context of forest research. Knowledge about the epigenetically regulated phenology of the vegetative buds of forest trees provides an idea regarding their ability to improve productivity, adaptability, and distribution potential during ongoing climate change [100], giving them an advantage over other plants.

Studies of methylation clearly show that it is involved in a number of key plant biological processes and is central to many plant developmental processes [106]. High-resolution genomic DNA methylation mapping studies based on the KEGG database have shown that methylated genes are involved in 118 metabolic pathways [114]. Many methylated genes encode proteins involved in chromatin structure and DNA synthesis, cell cycle regulation, nitrogen metabolism, fatty acid synthesis and elongation, starch and sugar metabolism, amino acid metabolism, protein metabolism, brassinosteroid biosynthesis, the tricarboxylic acid cycle pathway, hormone metabolism, and signal transduction pathways. These studies show that DNA methylation is involved in a wide range of biological processes [114], thus accounting for its great importance in development and environmental adaptation [80].

In plants, methylation-induced modifications may or may not be reversible but can be retained during cell division (mitosis and intragenerational transmission) in a memory process. An example of epigenetic memory is vernalization or transmission to the next generation during meiosis (identification of natural epivariants or artificially induced epivariants and epigenetic recombinant inbred lines) [1]. Furthermore, studies using epigenetically recombinant inbred lines (EpiRILs), where recombinant offspring are produced by crossing two parents with similar DNA sequences but strongly contrasting DNA methylation profiles, have shown that some of the DNA methylation variations are inherited in a Mendelian manner [61].

### 2.4. Epigenetic Modifications of Trees and Environmental Conditions—A Review of Existing Research and the Current State of Knowledge

A study on *Pinus radiata* D. Don [115] examining seedling tolerance to heat stress and priming, based on evaluations of the nuclear proteome and DNA methylation dynamics, identified proteins involved in epigenomically driven gene regulation. The authors believed that priming-induced epigenetic memory might drive the development of new methods to improve pine survival under extreme heat stress in the context of climate change. Facilitating tree acclimation through environmentally induced epigenetic memory has also been previously suggested for winter dormant shoot apical meristems (SAMs) of poplar field crops [116]. Understanding the mechanisms underlying phenotypic plasticity and stress memory in trees is extremely important in the context of rapid climate change. DNA methylation provides strong plasticity and modulates plant development, morphology, and physiology by controlling gene expression and transposable element (TE) mobility [116]. A type of phenotypic plasticity is epigenetic memory in the Norway spruce *Picea abies* (L.) H. Karst, and important factors for establishing this memory are DNA and histone methylation and sRNA [4,100]. There are three categories of stress-memory genes [117]. The first consists of “transcriptional memory” genes, in which stable transcriptional changes persist after a recovery period. The next contains genes called “epigenetic memory candidates”, in which stress-induced chromatin changes persist longer than the stimulus in the absence of transcriptional changes. The category following this comprises “delayed memory” genes, which are not immediately affected by the stress but receive and store the stress signal for a delayed response.

Memory-affecting climatic adaptation in Norway spruce is fixed at seed maturation during embryonic development and persists throughout the life of the offspring [100]. This mechanism allows for long-term adaptive phenotypic changes. The authors arrived at such conclusions following a transcriptional analysis of spruce seedlings from seeds of several full-sib families derived from different temperatures of embryogenesis (cold vs. warm) under long- and short-day conditions [100]. Epigenetic memory has evolutionarily important implications for trees growing in variable environments. In Norway spruce, which occurs over large areas, good adaptation to environmental conditions is provided by the epigenetic memory of temperature conditions during embryogenesis [113]. The large size of conifer genomes may also indicate a greater need for epigenetic regulation of chromatin structure and maintenance of chromatin in a “dormant” or non-transcriptional state until activated in response to a changing environment [4,113].

An intense decrease in global DNA methylation has been found in studies on winter-dormant shoot apical meristems of SAMs from natural populations of the black poplar *Populus nigra* L. in France subjected to summer drought [118]. To assess the extent of epigenetic changes, the authors examined common genetic parameters, such as narrow-sense heritability (h^2^), the phenotypic differentiation index (P_ST_), and the overall genetic differentiation index (F_ST_). As the results showed, a significant decrease in DNA methylation in these populations was associated with drought stress. Studies have also quite clearly shown that global DNA methylation, genetically and environmentally determined, can serve as a marker of natural population differentiation under drought stress [118], as well as performance or selection [79].

Findings for the distantly related Scots pine *Pinus sylvestris* L. populations located in northern and southern Finland suggest that DNA methylation and gene expression contribute to local adaptation in these populations and help the trees adapt to rapidly changing environmental conditions [5]. In megagametophytes, significant differences between populations have been detected in the expression levels of eleven adaptation-related genes. Similarly, in embryos, the expression levels of eight genes associated with adaptation differ significantly between populations. The study shows most genes with the strongest correlation to climate variables as having significantly different expressions between populations. These results indicate that DNA methylation plays an important role in ponderosa pine adaptation [5].

In contrast, a study of eucalyptus epigenetic variation [80] suggested that genetic background was the main driver of epigenetic variation. In that study, the DNA methylation patterns of four *Eucalyptus grandis* × *Eucalyptus urophylla* clones and one *Eucalyptus urophylla* S.T. Blake clone from two sites in Brazil contrasting in terms of water availability were compared. The aim was to relate these methylation patterns to environmental and growth traits. A stronger correlation was found between the detected DNA methylation and genetic background than between DNA methylation and location.

Analysis of the white poplar *Populus alba* L. DNA methylation profiles from vegetatively propagated populations [119] showed that environmental conditions strongly influence internal cytosine hemimethylation. Eighty-three samples of white poplar at different locations in Sardinia were investigated by MSAP. The analysis was performed on genomic DNA extracted from leaves at the same juvenile stage. The study showed that the genetic biodiversity of poplar is quite limited but is balanced by epigenetic interpopulation molecular variation. The results clearly showed that ramets of the same clone were differentially methylated according to geographical location. In poplar, epigenetic changes are frequent and occur more rapidly in response to environmental stimuli, confirming the molecular model of stress epigenetic memory for plant responses to stress leading to increased overall methylation levels induced by external stimuli [120].

The results obtained for natural populations of the endemic valley oak *Quercus lobata* Née of California occurring along the foothills of the Coastal and Sierra Nevada ranges provide further evidence for the role of methylation in the local adaptation or plasticity of plant responses to changing conditions [78]. Among 58 naturally occurring and species-wide samples of *Quercus lobata* collected across climatic gradients, 43 specific SMVs (single-methylation variants) significantly associated with one of four climatic variables (most associated with mean maximum temperature) were identified. Climate-related SMVs were mostly found near genes, some of which are involved in plant responses to the environment.

The relationship between environmental adaptation and DNA methylation has also been shown in studies on natural populations of the holm oak *Quercus ilex* L. of Mediterranean forests [121]. Methylation patterns and levels were assessed in individuals from control forest plots (in southern Catalonia, Spain) and in individuals experiencing drought stress (exposed to several years of drought at levels projected for decades to come). Drought-exposed plants had a percentage of hypermethylated loci lower than the control, while the percentage of fully methylated loci was significantly higher. These results also demonstrate that changes in DNA methylation contribute greatly to the ability of trees to rapidly acclimate to changing environmental conditions.

Cork oak *Quercus suber* L. is particularly tolerant to elevated temperatures, as indicated by its wide distribution in different climate zones growing in northern Africa and southern Europe (Mediterranean zone). In Europe, this oak is of great ecological and economic importance. The area of occurrence of the cork oak shows the large temperature variation to which this species had to adapt. The results of a study on adaptation to changing environmental conditions based on cork oak revealed high dynamics of DNA methylation in the temperature interval between 25 °C and 35 °C, with higher rates of de novo methylation than demethylation. In subsequent temperature intervals, there was a change in the baseline level of methylation, with stability up to 45 °C, followed by a marked increase in methylation [122]. This result suggests that under stress conditions, cork oak can rapidly regulate gene expression through DNA methylation to defend itself against the resulting stress conditions [122].

Mangroves are ecosystems found along tropical and subtropical coasts of the Americas and Africa. The mangrove species white mangrove *Laguncularia racemosa* (L.) Gaertn., native to salt marshes and riparian areas and thus subject to different environmental pressures, has shown abundant DNA methylation variation in studies, once again suggesting that epigenetic variation in natural populations plays an important role in adaptation to different environments [106]. Mangrove plant species, by virtue of their occurrence, must tolerate a wide range of environmental modifications and, therefore, represent an interesting natural system to study the correlation between DNA methylation levels, environmental conditions, and morphological traits [106]. As previously shown, the studied individuals (Sepetiba Bay, Rio de Janeiro, Brazil) showed remarkable genetic similarity, but morphological differences between individuals from different areas were surprisingly large, which the authors attributed to epigenetic variation. Simultaneously, they emphasized that DNA methylation and demethylation play an important role in the long-term adaptation of this species under different environmental conditions.

Most studies have shown that methylation levels vary among naturally occurring tree populations in different environments and suggest a link between methylation and local adaptation of the tree response to temperature changes (Table 1) [4,5,78,100,106,113,119,121,122].

Nevertheless, this is only the beginning of a full understanding of the function and operation of the epigenome. Many of the processes that occur during epigenetic modifications and the modifications themselves that have been most thoroughly understood in model plants (also mentioned above) have not been confirmed in studies of forest trees. Therefore, there is still a long way to go to fully understand the function of epigenetic modifications in trees in the context of both abiotic stresses and strictly global climate change.

## 3. Reprogramming Genes

Stress memory allows trees to respond faster and more vigorously to repeated stresses to increase stress tolerance or acquire the ability to avoid stress. Plants can “remember” certain past environmental experiences. Environmental conditions that fluctuate frequently can induce chromatin modifications and DNA methylation in various genes and consequently alter their activity. Environmentally induced chromatin modifications at some loci are heritable and can be passed on to the next generation, while others are reset at subsequent stages of growth and development [71]. Each successive generation of trees faces a different combination of environmental challenges. By losing or removing most environmental memories, plants provide offspring with the same adaptive start of earlier generations [71]. Many environmental memories last only for the lifetime of the plant, but some remembered events (somatic) can be passed on to future generations (transgenerational memory) [123]. To ensure the proper development of offspring, markers that accumulate at regulatory loci during growth and development or in response to environmental stimuli must be removed from gametes or embryos [59].

Chromatin modifications and remodeling and DNA methylation play important roles in complex plant–environment interactions and are indeed essential for adaptation [71,123,124,125,126]. We already have considerable knowledge concerning how trees receive environmental stimuli and the mechanisms by which environmental stress stimuli are transmitted to cellular signaling cascades and gene transcription networks [123]. However, it is important to understand exactly how environmental signals are transmitted to chromatin, how they exert chromatin changes, and how cell divisions transmit induced chromatin states [71].

Once we answer these questions, it may be possible to use the reprogramming mechanism to control and effect changes in long-lived plants such as trees. This knowledge would be useful in the context of adaptation to global climate change. Conversely, the occurrence of extreme weather events and repeated climate stress does not need to be fatal to many ecosystems, as previously predicted. Trees are able to adapt quickly to new conditions due to their sophisticated epigenetic mechanisms. The epigenetic memory of climate stress may contribute decisively to the adaptation of ecosystems to global climate change without human assistance.

Moreover, the constant trend that occurs with climate change differs from extreme, rapid weather changes, where plants do not have time to acclimatize [125]. Studies have shown that among the genes that are expressed during acclimation (e.g., to drought) are often genes of the LEA (late embryogenesis abundant) family [127], which are produced in plants during late embryogenesis and provide resistance to seed desiccation. Environmental stress memory may have a role in stabilizing plant communities exposed to extreme climatic events [125]. Studies suggest that the mechanism of environmental stress memory may rely on the accumulation of transcription factors or relevant proteins (that facilitate the rapid response after successive stress episodes), and this process may also involve epigenetic modifications (histone modifications, DNA methylation) that are inherited during mitosis or meiosis during cell divisions [74,128,129].

However, more studies are needed to determine whether the described potential of trees is sufficient to defend against the effects of climate change. Of particular interest, in the context of understanding the ecological response to abrupt climatic events, are the processes involved in the epigenetic inheritance of ecological stress memories that are beneficial to subsequent generations, and to what extent these processes interact with forest communities.

## 4. Editing the Epigenome of Trees, CRISPR/Cas9, and Other Molecular Tools

An important issue in the context of using epigenetics to improve knowledge about epigenetic modifications of trees and to improve trees to increase their adaptive potential is editing the epigenome. Several molecular tools are currently known to science, but whether it will be possible to apply them to tree research in the future remains to be seen.

Transcription, as the first step of gene expression, is subject to many regulations that also depend on epigenetic modifications (DNA methylation or histone modifications). This fact has contributed to the development of CRISPR-based epigenetic editors [130,131,132]. CRISPR (clustered regularly interspaced short palindromic repeat) is increasingly being used for targeted DNA methylation and demethylation, histone modifications and 3D DNA conformational changes, or epigenetic memory engineering.

Studies have shown that when attached to epigenetic effectors (EEs), CRISPR-dCas9 can also function as an epigenetic editing tool. The CRISPR-dCas9-EE module has been used to alter epigenetic features associated with various cancer tumor variants [133]. The authors believe that sgRNA-dCas9 can be used not only as part of a therapeutic strategy against cancer but can also become a versatile tool for epigenetic editing [133]. The CRISPR-Cas9 system consists of a CRISPR guide and a Cas9 effector (endonuclease). The CRISPR guide module directs DNA recognition, while the Cas effector module digests the target DNA. When the catalytic activity of the Cas9 protein is abolished by mutagenesis and a nuclease-free version of Cas9 (dCas9) is created, the CRISPR-dCas9 system can serve as a configurable genome-interacting device that binds to CRISPR-directed DNA sites [133]. Due to its uncomplicated effector structure, the type II CRISPR-Cas9 system has been adapted for genomic/epigenomic applications. All the structural features of CRISPR-Cas9 are shared by sgRNA-dCas9. Perhaps further research will allow the use of sgRNA-dCas9 to study tree traits in a broader context. The science of sgRNA-dCas9 has advanced significantly in recent years; nevertheless, many questions persist regarding the precise use of sgRNA-dCas9-EE and the factors that regulate off-target sgRNA-dCas9 binding, or regarding the molecular/cellular environment that promotes off-target effects [133]. The ability to accurately and efficiently design sgRNAs (taking into account target sequence features, epigenetic status, and chromatin context) and the efficiency of computational tools are also important. The technique could potentially be used to edit the genome of trees that are failing to cope with climate change but are species that are valuable to ecosystems and humans.

Although DNA methylation is a widely studied epigenetic modification, not the least of which relates to the important role it plays in gene expression and as a marker of molecular function, it is a process that poses a major challenge to researchers. Status control and manipulation of methylation are still fully unmanaged. An interesting method for controlling and editing DNA methylation is the CNAMS (clustered regularly interspaced palindromic repeats-Cas9-based near-infrared upconversion-activated DNA methylation editing system), which was designed for optogenetic editing of DNA methylation [134]. The CNAMS editing system consists of the fusion proteins of photosensitive CRY2PHR, the catalytic DNMT3A or TET1 domain, the CIBN fusion proteins, and catalytically inactive Cas9 (dCas9) [134]. This system can control DNA methylation editing in a spatial–temporal manner (in response to blue light) and provides the ability to remotely edit DNA methylation (after extending the spectral sensitivity from blue light to near-infrared (NIR) light). The authors of this method suggest the broad utility of CNAMS in functional studies of epigenetic regulation, which greatly expands the possibilities for epigenome control and research.

Recently, researchers at UC San Francisco and the Whitehead Institute described a remarkably novel tool that controls gene expression programs—a programmable CRISPRoff/CRISPRon epigenetic memory editor based on the CRISPR method [12]. The use of CRISPRoff (consisting of Cas9, among others) allows the turnoff of almost any gene in human cells without changing the genetic code. The deactivated gene remains inert in progeny cells for hundreds of generations until it is turned back on using CRISPRon (consisting of, among others, sgRNA). The technology is based on DNA methylation. When DNA is methylated, a methyl group is attached to it, causing nearby genes to be silenced. Gene expression control may occur for multiple genes simultaneously without any DNA damage, and vice versa [12]. CRISPRoff is highly specific and has a broad targeting window within gene promoters by identifying a single guide RNA (sgRNA) capable of silencing most genes, including those lacking canonical CpG islands (CGIs) [12]. The researchers designed a CRISPR-based programmable epigenome editor protein named CRISPRoff-V1 that consists of the ZNF10 KRAB, Dnmt3A (D3A), and Dnmt3L (D3L) protein domains and is linked to the catalytically inactive *S. pyogenes* dCas9. CRISPRoff can also be programmed by the orthogonal DNA-binding proteins dCas9 from *S. aureus* (dSauCas9) and dCas12a from *Lachnospiraceae* (dLbCas12a). To determine the ability of CRISPRoff to silence genes in the human genome and improve CRISPRoff activity, researchers have designed a sgRNA library (with over 20,000 protein-coding genes and containing approximately 1000 non-targeting sgRNAs) [12,135].

This newly discovered technology certainly still requires a great deal of research on its potential use in various fields. We do not know exactly how many genes are susceptible to stable versus metastable silencing or the regulatory features that dictate the stability of programmed epigenetic memory. What is certain, however, is that with CRISPRoff, scientists may have unprecedented control over the methylation process, which also changes the functional definition of DNA methylation itself [12].

The researchers believe that their discovery could become a promising tool for treating rare genetic disorders caused by the activity of a single defective copy of a gene, such as Marfan syndrome, Job syndrome, immune system disorders, and cancer [12]. Similar to most other previously discovered research techniques, CRISPRoff/CRISPRon will hopefully also become another tool in the study of plants, including forest trees. The epigenome plays a key role in many diseases, from viral infections to cancer, and CRISPRoff technology provides a space to research new and improved epigenetic therapies for human cells. Similar hopes can be placed on plant research. Because of the important biological functions performed by the epigenome in the plant cell, the possible application of this technology could also become groundbreaking.

CRISPR-Cas9 can be used for gene silencing when coupled with DNA methyltransferase and for gene activation when coupled with the catalytic domain of TET1 [130,131,136]. The CRISPR-based dCas9-SunTag system was used to target gene activation and DNA methylation in *Arabidopsis thaliana*, contributing to the induction of FWA promoter methylation and the early flowering phenotype [19]. The researchers modified the SunTag system to recruit multiple copies of the methylation effector or VP64 to different loci. To accomplish this goal, the catalytic domain of *Nicotiana tabacum* L. DRM methyltransferase was initially described as a methylation effector. As shown in the SunTag study, NtDRMcd effectively directed methylation to specific loci. The FWA locus in the methylated state is meiotically heritable for multiple generations in the absence of targeting transgenes [19]. This study suggested that the SunTag-VP64 system is a valuable tool for epigenome manipulation in plants and can be used to study the epigenetic regulation of methylated loci without changing the global level of DNA methylation.

Induction of targeted DNA methylation to modify gene expression can also be accomplished by creating a platform for precise editing of the epigenome using epigenome modifiers (DEMs) [137]. DEMs combine in a single molecule a DNA-binding domain based on highly specific transcription activator-like effectors (TALEs) and several effector domains capable of inducing DNA methylation and local chromatin rearrangement to silence the expression of target genes [137]. Nevertheless, the main advantage of CRISPR-Cas9 over other gene-editing tools is the ability to use sgRNA—which is easier to design—without the need to design specific DNA-binding proteins [138].

Next-generation sequencing technology, under pressure from climate change and the associated need to study epigenomic diversity [139,140,141,142,143], has been initiated and is successively advancing the study of tree epigenomics [1,144]. Genomics is an important component in uncovering epigenetic modifications for environmental adaptations and developing innovative interventions for maintaining forest adaptive capacity. Nevertheless, to put these capabilities into practice, reference sequences of the (epi)genomes of forest trees need to be established [1]. In addition, a unified interpretation of the obtained sequencing results is needed in the context of adaptation [140,144,145]. Despite the existence of data surveys (https://www.hardwoodgenomics.org/ or https://phytozome-next.jgi.doe.gov/, accessed on 19 July 2021), there is a need for comprehensive databases on forest trees and the possibility of using them on an open-access basis, following the example of the Human Genome Project. This approach is one of the main factors accelerating scientific progress in research. When databases of forest trees are freely shared and reach a wide range of experts, large-scale sequencing will be most effective [145].

In addition, the growing knowledge of m5C DNA glycosylases may also be relevant to the growing field of epigenetic editing [146]. Some studies support the possibility of using m5C DNA glycosylases as molecular tools to modify cellular methylomes. Perhaps further studies will provide new opportunities concerning the molecular biology of active plant DNA demethylation in trees and its role in physiological processes and supply knowledge on new applications in developing epigenetic technologies.

Epigenome editing techniques are one of the most interesting molecular tools, in the context of intervening and helping forest ecosystems if, in a changing climate, trees need it. They would be worthy of broader interest in the context of forest tree epigenome editing.

## 5. Conclusions

Forests and trees have an amazing ability to survive. In this context, there is a need to understand the extent to which epigenetics play a role in resilience and plastic adaptation to the environment. The answer to this question is crucial in predicting how trees may behave under ongoing, continuous climate change associated with extreme events, such as drought, temperature extremes, and insect defoliation.

Concurrently, our knowledge is still limited. The frequency of environmentally induced epigenetic changes is an open question. Are they episodic processes that destabilize epigenetic homeostasis [25]? Do they occur frequently and appropriately to stimulate environmental stimuli, providing an adaptive response each time [73]? Furthermore, the mechanisms of siRNA-mediated methylation control and the importance of gene expression under normal or adverse growth conditions are poorly studied [13], in addition to epigenetic modifications of individuals in relation to populations—the commitment and speed (within populations) at which epigenetic variants may evolve in response to stress conditions so that the population can survive under rapid environmental change [73]. The exact number of tree species and the forests they form in response to climate change is also unknown.

Rapid changes in environmental conditions, including global climate change, require new plant engineering approaches that will rely on the control of methylation and activation of transposons as factors in transforming pathways and gene regulatory networks to induce new traits and physiological resilience [73].

## Figures and Tables

**Figure 1 ijms-22-13524-f001:**
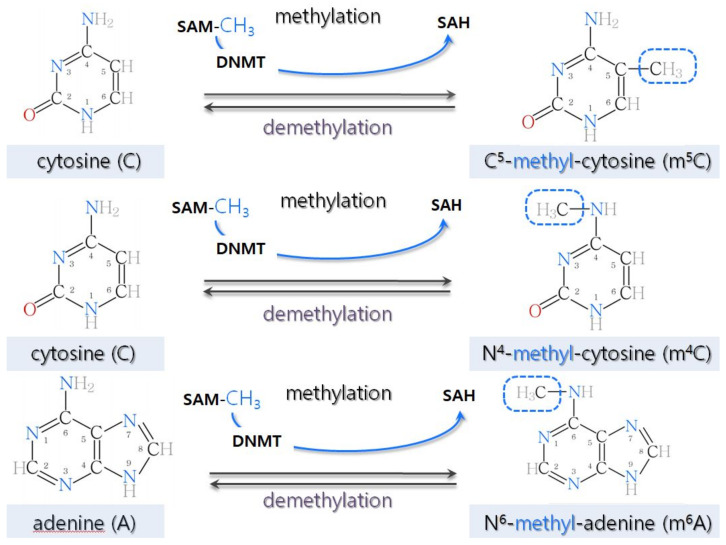
Structural models of substrates and products of DNA methylation. Cytosine (CYT or C) is a nitrogenous base from the pyrimidine group that forms via three hydrogen bonds in double-stranded nucleic acids and a complementary pair with guanine (GUA or G). Adenine (ADE or A) is a nitrogenous base from the purine group that, by means of two hydrogen bonds in double-stranded nucleic acids, forms a complementary pair with thymine (THY or T in DNA) or uracil (URA or U in RNA). In methylation, the methyl donor is most often S-adenosyl-L-methionine (SAM), yielding S-adenosyl homocysteine (SAH), which is an inhibitor of methyltransferases; the entire process is catalyzed by DNA methyltransferases (DNMT) and is cytosine- and adenine-specific. DNMTs transfer methyl groups to the appropriate positions on the rings of nitrogenous bases: into carbon (at position 5 of the cytosine ring) and amino groups (at position 4 of the cytosine ring and position 6 of the adenine ring) [6].

**Figure 2 ijms-22-13524-f002:**
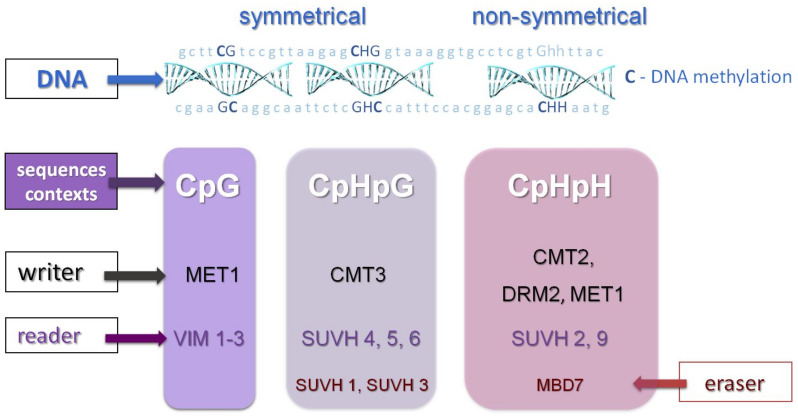
Maintenance methylation in plants. DNA METYLOTRANSFERASE 1 (MET1) is required to maintain DNA methylation in the context of CpG dinucleotides. Methylation variant (VIM) proteins are required here—VIM proteins 1–3 function in transcriptional regulation through their role in the MET1 DNA methylation pathway. Maintenance of DNA methylation in the context of CpHpG requires CHROMOMETHYLASE 3 (CMT3) activity. In the context of CpHpH, this activity is mainly controlled by CHROMETYLASE 2 (CMT2) but also by DOMAINS REARRANGED METHYLTRANSFERASE 2 (DRM2) through the RNA-directed DNA methylation (RdDM) pathway. Plant-specific chromometallases are both “readers” of histone methylation and “writers” of DNA methylation. They interact with SUVH histone methyltransferases. MBD domain proteins in plants can act both as “readers” (MBD5, 6 for CpG context) and “erasers” of DNA methylation (MBD7).

**Figure 3 ijms-22-13524-f003:**
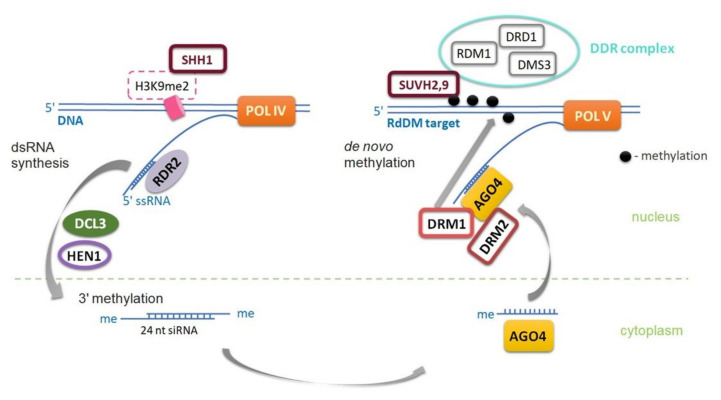
RNA-directed DNA methylation (RdDM) pathway in plants. The first step in de novo DNA methylation pattern formation is the RNA-dependent DNA methylation (RdDM) pathway, which relies on the specialized plant-specific RNA polymerases POL IV and POL V (in all three sequence contexts). RNA POLYMERASE IV (POL IV) cuts into short (26–45 nt), single-stranded RNAs (ssRNAs) that serve as a substrate for RNA-dependent RNA POLYMERASE 2 (RDR2). RDR2, together with POLI V, converts them into double-stranded RNA (dsRNA). The dsRNAs are then converted into 24-nucleotide small interfering RNAs (siRNAs) by DICER-LIKE 3 (**DCL3**), methylated at their 3′-end by HUA ENHANCER 1 (HEN1), and recruited by ARGONAUTE 4 (AGO4) or other ARGONAUTE proteins (AGO6 and AGO9). AGO4-siRNA complexes, interacting with POL V, then recruit DOMAINS REARRANGED METHYLTRANSFERASE 1 and 2 (DRM1, DRM2) DNA methyltransferases to the target DNA. POL V can be recruited by indirectly interacting with histone methyltransferases SU(VAR)3–9 homolog 2 (SUVH2 and SUVH9), which act as de novo methylation “readers” through interaction with the DDR complex (DRD1, DMS3, RDM1). The DDR complex consists of DEFECTIVE IN MERISTEM SILENCING 3 (DMS3), DEFECTIVE IN RNA-DIRECTED DNA METHYLATION 1 (DRD1), and RNA-DIRECTED DNA METHYLATION 1 (RDM1). The DNA methyl-readers SUVH2 and SUVH9, along with the DDR complex, are required for POL V recruitment to chromatin. The SAWADEE HOMEODOMAIN HOMOLOG 1 (SHH1) histone reader is required for POL IV association to chromatin. AGO4-siRNA complexes are then targeted to transcripts generated by POL V and recruit DOMAINS REARRANGED METHYLTRANSFERASE (DRM1, DRM2) DNA methyltransferases to the target DNA [22,23].

**Table 1 ijms-22-13524-t001:** Studies of epigenetic modifications of forest trees under environmental stress.

Species	Type of Modification	Stress Condition	Method	Literature
*Pinus radiata*	changes in tissue DNA methylation dynamics	heat stress and priming	quantification of nuclear proteins by nLC-MS/MS	[115]
*Picea abies*	epigenetic memory—increase in overall DNA methylation levels induced by external stimuli	climate adaptation	expression analysis of 32 genes by qRT-PCR	[4,100,113]
*Pinus nigra*	decrease in global DNA methylation	drought stress	genome-wide SNPs	[118]
*Pinus sylvestris*	effect of DNA methylation on expression of 11 genes	environmental adaptation	DNA global methylation, GC/MS	[5]
*Eucalyptus grandis* × *Eucalyptus urophylla* and *Eucalyptus urophylla*	a stronger correlation between DNA methylation and genetic background than between DNA methylation and location	environment and growth characteristics	MS-DArT-seq, methyl Sensitive DArT-seq sequencing	[80]
*Populus alba* L.	DNA methylation dynamics—changes in methylation in relation to geographical location	climate adaptation	MSAP, methylation-sensitive amplified polymorphism	[119]
*Quercus lobata*	43 single-methylation variants were significantly associated with climatic factors, such as mean maximum temperature	climate adaptation	RRBS, reduced-representation bisulphite sequencing	[78]
*Quercus ilex*	DNA methylation dynamics—the percentage of fully methylated loci was significantly higher	heat stress	MSAP, methylation-sensitive amplified polymorphism	[121]
*Quercus suber*	increase in DNA methylation at higher tepmeratures	heat stress	MS-RAPD, methylation-sensitive random-amplified polymorphic DNA	[122]
*Laguncularia racemosa*	variability of DNA methylation relative to populations—for all MSAP markers, identified 67 loci with CpG-methylation, 116 non-methylated loci and 26 hemimethylated loci	climate adaptation	MSAP, methylation-sensitive amplified polymorphism	[106]

## Data Availability

Not applicable.

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
