# Peer review of "Can Forest Trees Cope with Climate Change?—Effects of DNA Methylation on Gene Expression and Adaptation to Environmental Change"

_ijms, 2021, doi:10.3390/ijms222413524_

Round 1
Reviewer 1 Report
In this manuscript, the authors summarise findings about epigenetic modifications in trees, and their contribution to the regulation of gene expression and adaptation to environmental changes. There is also recent information about the epigenome editing tools, such as CRISPR/Cas9 and other molecular tools.
Overall, the manuscript is well written, up to date and illustrated by three original figures and a table. The main critique I have is that some parts seem to be a bit disconnected from the main manuscript topic. The title says that the main focus should be put on forest trees and the effects of DNA methylation but a large part of the manuscript deals with general questions. One example is the information provided in the second and the following paragraphs of part 2.2 “Forest trees – ecosystems important to humans”. I really appreciate the collected information but it is too lengthy and detailed. The last paragraph (page 7) provides more information on Arabidopsis and only the end of this paragraph (page 8) focuses on the studies related to forest tree epigenetics. I suggest to make part 2.2 more focused, concise and linked to the manuscript topic.
Detailed comments:
Title: Please give more attention to the manuscript title - “Climate Change”, “Environmental Change”...
Affiliation: the same affiliation is listed twice – correct according to the journal instructions.
Page 1 - Could you please rephrase the first sentence of the Introduction - since you are talking about something that is known, change the verb tense from past to present/present perfect. Also the first two sentences of the second paragraph seem to be slightly confusing. Please rephrase.
Page 2 – subtitle 2 – correct DNa format. What do you mean by DNA methylation is one of “the most permanent processes”? The sentence after - DNA methylation not always leads to silencing of gene expression.
References do not follow the journal’s instructions throughout the text. They must be numbered in order of appearance in the text, and the reference numbers should be placed in square brackets and listed at the end of the manuscript.
The first paragraph of part 2.2. “Forest trees – ecosystems important to humans” (page 5-6) again focuses on changes in chromatin structure and adaptive epigenetic modifications, and to some extent repeats previous statements. I would suggest to rephrase/shorten and move this part above. Part 2.1 could be expanded and potentially retitled. Part 2.2. can directly start with the second paragraph dealing with forests.
In 2.3 “Effects of DNA methylation on adaptations of forest trees” is provided information about the techniques to analyse DNA methylation (page 8-9). Since part 4. “Editing the epigenome of trees” focuses entirely on methodology, you can move this piece of information here.
Introduce the abbreviation for “DArT-seq” (page 9)
The abbreviation of SAMs is already introduced and wrongly repeated - “shoot apical meristems of SAMs” (page 10).
Replace “5. Concludions” with “5. Conclusions” (page 17). This part could also be more concise.
Overall, the manuscript presents a nice overview of the current knowledge of epigenetic modifications in trees and recent technologies, and should be considered for publication in IJMS, once all the issues are solved.
Author Response
Dear Sir/Madam,
Thank you for your good feedback on our manuscript and your detailed comments, which also in our opinion enhance the value of the text. The comments have been incorporated into the manuscript and we hope that we have interpreted all of them correctly.
Best regards
Ewelina A. Klupczyńska

Reviewer 2 Report
The manuscript provides a depth-insight into current knowledge about DNA methylation in trees in the context of global climate change, and the potential of molecular tools for study epigenetics. The manuscript is excellent prepared and concentrate all important information that are systematically presented to the reader.
I have no further suggestions for its improvement.
Author Response
Dear Sir/Madam,
Thank you very much for your favorable review of our manuscript. We are grateful for the very positive review.
Best regards
Ewelina A. Klupczyńska